# Evaluating Supplementary Water Methodology with Saturated Soil Irrigation for Yield and Water Productivity Improvement in Semi-Arid Rainfed Rice System, Burkina Faso

**Aimé Sévérin Kima [1,2,*], Etienne Kima [3], Bernard Bacyé [4], Paule A. W. Ouédraogo [4], Ousmane Traore [5], Seydou Traore [6,7], Hervé Nandkangré [8], Wen-Guey Chung [9] and Yu-Min Wang [9]** 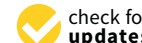

[1] Environment and Agriculture Research and Training Center of Kamboinsé, Institute of Environment, and Agricultural Research, Ouagadougou 01 01 B.P. 476, Burkina Faso

[2] Department of Tropical Agriculture and International Cooperation, National Pintung University of Science and Technology, 1 Hseuh Fu Rd., Neipu Hsing 91201, Pingtung, Taiwan

[3] Ministry of Agriculture, Direction of Irrigation, Ouagadougou 03 03 BP 7123, Burkina Faso; tindakima@yahoo.fr

[4] Institute of Rural Development, Nazi Boni University, Bobo-Dioulasso 01 01 BP 1091, Burkina Faso; bbacye@gmail.com (B.B.); paulealiceo@gmail.com (P.A.W.O.)

[5] School of Economics, Anhui University Hefei, No. 3, Feixi Road, Hefei 230039, China; ousmantra711@gmail.com

[6] 207B Scoates Hall 2117, Department of Biological & Agricultural Engineering, Texas A & M University, College Station, TX 77843, USA; se73traore@gmail.com

[7] Global Development & Innovative Services, 13801 Walsingham Road, #A-119 Largo, Florida, FL 33774, USA

[8] Polytechnic Centre of Tenkodogo, University of Ouaga 2, Ouagadougou 12 12 BP 417, Burkina Faso; hervenankangre@yahoo.fr

[9] Department of Civil Engineering, National Pintung University of Science and Technology, 1 Hseuh Fu Rd., Neipu Hsing 91201, Pingtung, Taiwan; chungwg@mail.npust.edu.tw (W.-G.C.); wangym@mail.npust.edu.tw (Y.-M.W.)

[*] Correspondence: aimeseverinkima@yahoo.fr; Tel.: +226-7029-4857 or +226-6655-9717

**Abstract:** Saturated soil irrigation (SSI) has been reported as a great technique that increases water productivity in fully irrigated rice cultivation. However, this technique should be employed in a dry prone area where rainfalls fail to fulfill rice water requirements and fill up reservoirs for sufficient irrigation. Therefore, an experiment was conducted to identify the most effective complementary irrigation that restores soil moisture to saturation and increases water productivity with fewer yield expenses. The study used a complete randomized blocks design with four replications and four soil saturation treatments: $Sat_{200\%}$ (farmer practice), $Sat_{160\%}$, $Sat_{120\%}$, and $Sat_{80\%}$. In $Sat_{160\%}$, $Sat_{120\%}$, and $Sat_{80\%}$ irrigation was applied once a week in the absence of rainfall. In $Sat_{200\%}$, water was daily applied except the day of rain. The results showed that reductions in soil saturation decreased plant height, tillers, and leaves number by 10%, 18%, and 14%, respectively. Yield and components were reduced between 26.09% and 4.8%. Weekly soil saturation at 120% exhibited greater irrigation productivity ($0.69 \text{ kg/m}^3$), rainwater productivity ($1.02 \text{ kg/m}^3$), and water-saving (90.53%) with less production penalty ($5 \times 10^{-3} \text{ kg/m}^3$). We advocate that saturated soil irrigation should be recommended in the rainfed rice system as a drought mitigation measure in semi-arid conditions.

**Keywords:** complementary irrigation; water productivity; saturated soil irrigation; semi-arid conditions; rainfed rice system

## 1. Introduction

Concerns are mounting worldwide about producing more rice per drop of water to feed rapidly growing populations, particularly in semi-arid and arid conditions where rainfalls have failed to fulfill crop water requirement. Improving yield per unit of water is a key concern in the face of the combined impacts of a growing population and food demand [1,2], and these concerns are particularly acute in rainfed agricultural systems, which often have lower productivity [3]. In countries located in these dry prone regions, like Burkina Faso, rainfed rice production is constrained by rainwater shortage, and complementary irrigation is crucial to maintaining and/or increasing yield and production to meet fast-growing demand. The country's agriculture is rainfed, and crops are subjected to the low and inadequate rainwater supply. The high fluctuation of rainfall in addition to several drought spells in rainy season lead to poor rice yields and low production growth, while an increase of 3% rice is required every year [4]. A key strategy is to minimize risk for dry spell induced crop failures, which requires an emphasis on supplemental irrigation [5].

Drought is the major climate constraint negatively affecting local community in the semi-arid zone of Burkina Faso [6], and yield deficit is caused by a decrease of rainfalls [7,8]. In this country, the rainfed agriculture is subjected to climate variability, and recurrent droughts spells have become the most important threat to ensuring food security [9]. In dry areas, water is the major constraint for increasing agricultural productivity [10], and supplementary water is required [11]. In semi-arid and arid regions, drought stress is one of the most severe environmental stresses, causing serious reduction in growth, quantity, and quality in many crops [12]. In supplementary irrigation, additional water is applied to fill the gap between rainfalls and to address drought occurrence. Supplementary irrigation application during a period when rainfall cannot provide sufficient water for plant growth should improve or stabilize yield [13].

Recently, the government of Burkina Faso has decided to support supplementary irrigation for ensuring rice production. The supplemental irrigation method is widely promoted, and related projects are financially supported by the country's government [14]. Extension service workers and farmers resort to supplementary irrigation without much attention to the applied water depth and effective rainfall amount and pattern. Supplementary irrigation is applied based only on water availability. The effect of water management on irrigation schemes is not "strongly" considered [15]. Indeed, applying water regardless of rainfall and suitable water depth leads to the use of large amounts of water and low productivity. Moreover, in the country, rice is exclusively grown under surface irrigation in which a high quantity of water is lost at each application of irrigation. In Burkina Faso, surface irrigation is the most dominant rice irrigation system, but the efficiency of water application is lower than 20% [16]. In the surface irrigation of rice fields, a large amount of water (around 1.5 million m$^3$ per site) is lost every year through seepage and deep percolation [17]. In such a context, the great challenge is to cope with water scarcity by improving water use efficiency. Evaluating supplementary irrigation alongside rainfall occurrence should help to find out the suitable water depth for increasing productivity. The efficiency of supplementary irrigation is driven by both an effective amount of rain at different growth stages and by a supplementary water depth. Kima et al. (2015) [18] mentioned that irrigation should always consider rainfall occurrence and suitable water depth for improving the effectiveness of the irrigation system. The effectiveness of supplementary irrigation depends on water losses that occur during and after the irrigation. Applying high ponded water depth, regardless of rainfall, may increase water losses through percolation and/or surface runoff.

Dwindling agricultural water supply coupled with increasing drought frequency and the uncertainty of rainfalls mean that irrigation systems need to improve water use efficiency or water productivity. The water crisis in semi-arid areas accentuates the need for improving water use efficiency in rainfed agriculture to build resilience for coping with future water related risks and uncertainties [5]. Since rainfall failed to fill up reservoirs for sufficient supplementary irrigation, maximizing water productivity should be the better strategy for rainfed farming systems. It is crucial to improve water use efficiency in agriculture, particularly in arid and semi-arid areas where the need

is greatest [19]. In this context, farm water management techniques and practices have been generated and promoted to sustain both rainfed and irrigated crop production, especially of rice, which is the most important water consumer. Among these techniques, saturated soil irrigation (SSI) has been widely reported as a technique that can save a lot of water whilst increasing water productivity with less yield penalty [20–24].

In SSI, the soil moisture is kept close to saturation as much as possible, thereby reducing ponded water and then decreasing seepage and percolation flows. However, the drawback of this technique is that it requires too much time for daily irrigation, and it is exclusively employed in full irrigation [25,26]. This technique should be employed as complementary irrigation to overcome rainfall uncertainty and rainwater shortage if the suitable water depth is defined at a reasonable irrigation interval.

Since rainfall is the main source of water for the rainfed rice crop, supplementary irrigation is not applied to create soil water-stress-free conditions during the cropping season but to ensure that optimum water is available, especially during high water need periods, for optimizing water productivity and maximizing yield [13]. Employing SSI under supplemental irrigation methods may reduce water losses with positive impact on productivity. Keeping soil moisture close to saturation may optimize root water uptake while reducing surface runoff and deep percolation. The irrigation interval and water depth should be adjusted to maintain soil moisture up to stress level as long as possible so that plants can maximize water absorption. Optimization of water uptake by plants necessitates changes in the application of irrigation water and the interval between consecutive irrigation [27]. Irrigation may be kept at a defined interval and water adjusted accordingly. In tropical climate conditions, it was found that seven days interval can be adopted and water depth adjusted accordingly [26]. To our knowledge, such an approach is little documented and has never been employed in semi-arid rainfed cropping system. This gave us an impetus for the present experiment.

Therefore, finding out the effective water depth for optimizing water consumption while improving yield should help in alleviating drought effects and scheduling proper supplementary irrigation for increasing farm profitability. Such an approach may reduce water consumption and maintain high productivity. The current study aims to evaluate supplemental irrigation for determining a suitable water depth that may improve water use efficiency through the SSI technique, which in turn will increase water productivity and farming output in dry climate conditions.

## 2. Material and Methods

### 2.1. Site Description and Trial Design

The experiment was conducted in the 2019 rainy season from July to November at the Bagré irrigated site (11.30° N and 0.25° W) located in a semi-arid region in eastern Burkina Faso. The soil was sandy and loamy with a wilting point of 12.0% volume, a field capacity of 21.5% volume, a saturation of 33.3% volume, a bulk density 1.7 g cm$^3$, and infiltration rate of 25 mmh$^{-1}$. The experimental design was a randomized complete block with four replications and four soil saturation treatments (Sat$_{200\%}$, Sat$_{160\%}$, Sat$_{120\%}$, and Sat$_{80\%}$). Each plot was 3 m × 3 m, and the soil bed height was 0.3 m. The spacing between plots and between blocks was 2 m.

### 2.2. Land Preparation and Crop Establishment

Wetland preparation was done on 5 August 2019 through manual tillage at 10 cm depth. Bunds were made at 20 cm height and well compacted. A week later, plots were soaked with 5 cm ponded water depth and manually leveled. On 13 August, as is usually done by farmers, twenty-five-day old seedlings of the Orylux 6 variety were manually transplanted at 25 cm × 25 cm hills spacing. Orylux 6 is a short duration (105 days) and drought tolerant variety with a yield ranging from 3 to 6 t/ha. Two weeks after transplanting, fertilizer (NPK), with a ratio of 14:23:14, was applied at the rate of 200 kg/ha. Urea (46%) was applied two times at rates of 50 kg/ha and 100 kg/ha through active tillering

and panicle initiation, respectively. Pesticides were applied on demand to control specific rice pests. Weeds were frequently and manually removed during the experiment.

## 2.3. Irrigation Management and Soil Moisture Analysis

After transplanting, irrigation was applied at soil saturation when needed for seedlings adaptation. Two weeks after transplanting, first water treatments were applied to reach the desired soil saturation percentage defined using the equation described by Ekren et al. (2012) [28].

For a rice crop, maximum roots are found at the depth of 30 cm [29], so that depth was considered in the equation.

In this study, undisturbed soil properties were used for the calculation of applied water depth. In fact, in the Saturated Soil Irrigation (SSI) technique, applied water is calculated based on undisturbed soil properties [28]. The aim of the approach is to keep soil moisture as close to saturation as possible [25].

Plastic tubes were buried into the soil in all plots so that 5 cm, 7.5 cm, 10 cm, and 12.5 cm of their length protruded above the soil surface, and irrigation water was conveyed through the furrow and applied accordingly to reach $Sat_{80\%}$, $Sat_{120\%}$, $Sat_{160\%}$, and $Sat_{200\%}$, respectively. The interval of 40% was chosen to practically ease the implementation with furrow irrigation. In treatments $Sat_{160\%}$, $Sat_{120\%}$, and $Sat_{80\%}$, irrigation was applied once a week when there was no rainfall, but it was postponed to the following week when rain occurred. In $Sat_{200\%}$ treatment, irrigation was applied daily except when rain occurred and was usually done by the farmer. Water application in all treatments stopped two weeks before harvest on 21 October.

During land leveling, water tubes with 45 cm length and a diameter of 15 cm were placed on the side of the plot close to the bund at a distance of 1 m to observe soil water fluctuation. Holes of 0.5 cm diameter, spaced 2 cm away from one another, were drilled into the bottom thirty lengths of the water tubes. Tubes were buried 30 cm vertically into the soil, and water variation was daily recorded at the same time using a ruler. The soil water level in tubes was determined by subtracting reading (cm) from the tubes' length.

Soil moisture stress limit was determined based on the accepted water depletion limit for rice crops to analyze water fluctuation. According to Bouman et al. (2007) [25], the stress limit was 25 cm below soil surface in arid and semi-arid conditions.

## 2.4. Assessment of Growth Parameters

After transplanting, 12 hills were randomly selected through diagonals and medians in each replicate for growth parameters assessment. Measurements (plant height, tillers, and leaves number) were done from each of the 12 selected hills individually with active tillering and panicle initiation. Plant height was measured from the base to the tip of the highest leaf using a ruler. Tillers and leaves from each hill were individually counted. From the first panicle emergence until 50% of the farm was headed, daily panicles were counted from each plot for heading rate analysis.

## 2.5. Assessment of Yield Components and Yield

At harvest, panicles from the selected hill were cut from the base and separated from the straw. Tillers and panicles were manually counted to determine the number per hill. Panicles were individually measured and three length classes were determined per hill according to the statistical method described by [14]. Three panicles were randomly picked (one from each length class) from each hill to measure the length and the weight.

Selected panicles were hand-threshed, and grains were weighted using a scale. The grain number was manually counted. Panicles from each hill were also threshed and grain weight per hill was obtained. Grain yield (kg/ha) and straw weight (kg/ha) were determined using the equation described by Kima et al. (2015) [18].

One sample of 1000 grains was taken from grains of each selected hill and weighted to obtained 1000 grains weight.

## 2.6. Water Productivity

Complementary water productivity ($CWP_i$), rainwater productivity ($RWP_i$), and Total water productivity ($TWP_i$) were calculated using equations defined by Geerts and Raes (2009) [30].

Production losses were determined considering the yield in the highest water treatment as a reference, and the "impact of water savings" was defined as the grain production lost by saving one unit of irrigation water. "Impact of water savings" (IWS) was obtained through the equation below [26]:

$$IWS = \frac{Y_{losses}}{IR_{saved}} \tag{1}$$

where IWS represents the impact of water-saving ($kg/m^3$), $Y_{losses}$ is the yield losses (kg/ha), and $IR_{saved}$ is the irrigation water saved ($m^3$/ha).

## 2.7. Statistical Analysis

The statistical analysis was done using SPSS 18 software (IBM, Armonk, New York, NY, USA). The data were subjected to the analysis of variance. The significance of the effect of the treatment was determined using a F-test, and means were separated through Duncan's test at 95% confidence.

## 3. Results

### 3.1. Environment Conditions

Figure 1 presents monthly effective rainfall, radiation, and crop evapotranspiration. Weather data were obtained from a national meteorological station located at Bagré. Rice evapotranspiration was estimated using a standard FAO Penman-Monteith formula by running CROPWAT 8.0. Heavy rain and high rain frequency were observed from July to September. From October to the end of March, no rain was registered. Light rainfalls were recorded from April to June. The highest radiation was registered from March to June, while the lowest values were observed in December and January.

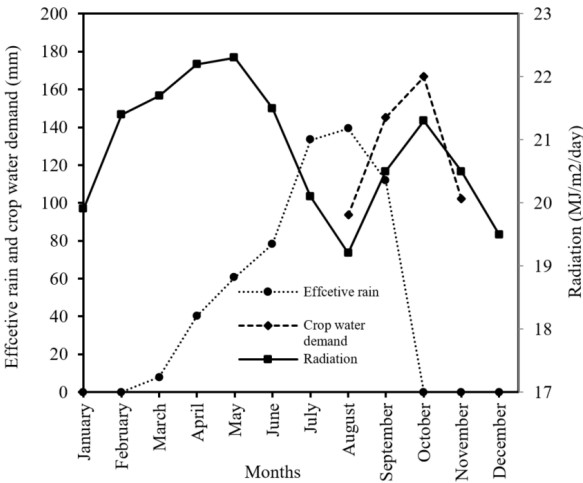

**Figure 1.** Sketch of monthly effective rain, crop water demand, and radiation in the production area during the cropping period.

The rainfall pattern and radiation trend showed that the environment was characterized by three main seasons (one rainy season and two dry seasons). The rainy season starts in July and ends in October, followed by a cold and dry season from November to February, and a hot dry season from

March to June. During the cropping period (13 August to 3 November), rainfalls varied from 400 to $0 \text{ m}^3 \text{ ha}^{-1}$. Highest rainfall values (250, 300, and 400 $\text{m}^3 \text{ ha}^{-1}$) were recorded at the vegetative stage with a total of 3030 $\text{m}^3 \text{ ha}^{-1}$, while no rain was registered at reproductive and repining stages (from 27 September to 3 November). During the cropping period, only 65% of rice water needed was covered by rainfalls. However, crop evapotranspiration was not fulfilled at a sensitive stage.

### 3.2. Soil Water Trend

Soil moisture fluctuation at different growth stages is presented in Figure 2. Soil water content varied according to rainfall occurrence and the saturation levels and was significantly different among treatments. After rainfall events and/or irrigation application, soil moisture increased up to a maximum level followed by a decrease in all treatments. When rain occurred or when irrigation was applied, soil water content was almost similar in all treatments. In the absence of rainfall, soil moisture decrease was different in treatments and depended on the amount of supplemental water.

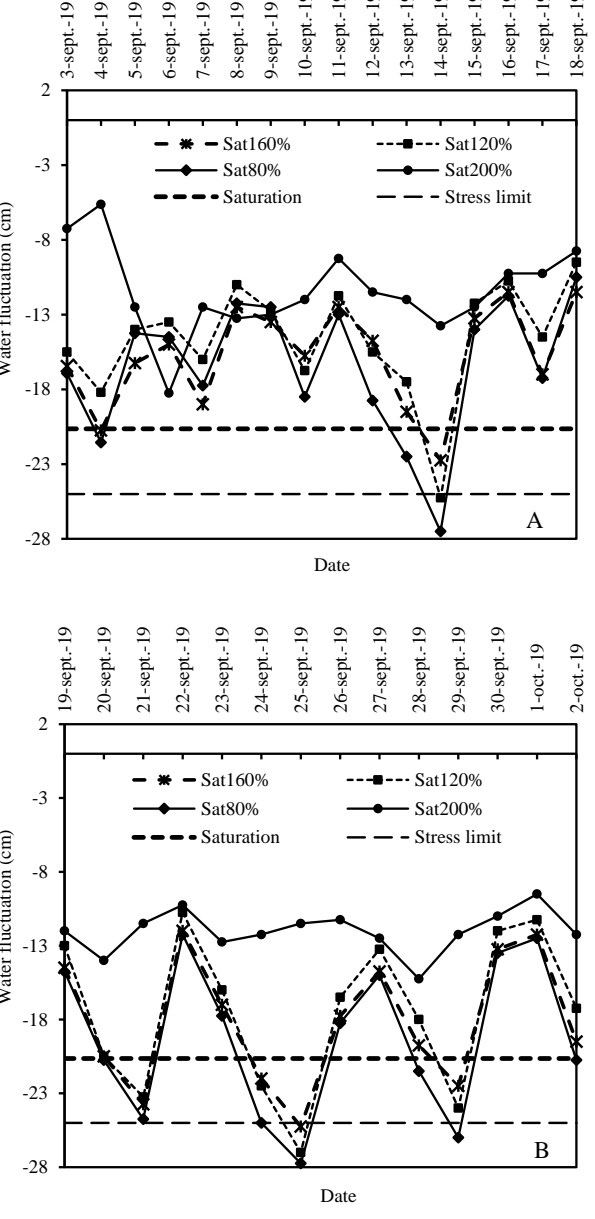

**Figure 2.** *Cont.*

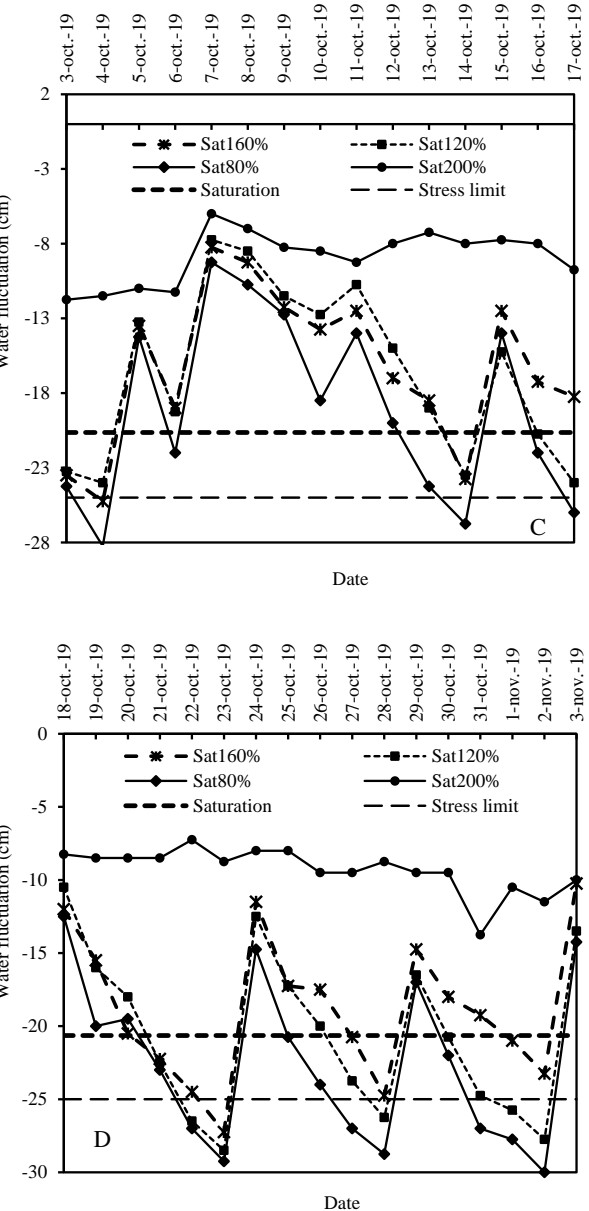

**Figure 2.** Comparison of the soil water variations in water tubes from the initial stage to tillering (**A**), from tillering to panicle initiation (**B**), from panicle initiation to heading (**C**), and from heading to harvest (**D**).

At all growth stages, soil moisture was globally higher in $Sat_{200\%}$ and stood up to saturation level, although lower values were recorded on 6, 8, and 9 September, maybe due to deep percolation losses because of high ponded water depth. From the initial stage to tillering (Figure 2A), moisture dropped down to the stress level one time in $Sat_{80\%}$ and $Sat_{120\%}$, while no stress was observed in $Sat_{160\%}$. A short duration of stress was observed in $Sat_{80\%}$ and $Sat_{120\%}$. From tillering to panicle initiation (Figure 2B), soil moisture settled down to the stress limit two times in $Sat_{80\%}$ and one time in both $Sat_{160\%}$ and $Sat_{120\%}$. Water stress was relatively sharper in $Sat_{80\%}$ compared with $Sat_{160\%}$ and $Sat_{120\%}$ From panicle initiation to heading (Figure 2C), soil water content was generally lower in $Sat_{80\%}$ compared with $Sat_{160\%}$ and $Sat_{120\%}$. Soil moisture settled in the stress zone three times in $Sat_{80\%}$ and one time in $Sat_{160\%}$, while no stress was observed in $Sat_{120\%}$ From heading to harvest (Figure 2D), stress was observed seven times in $Sat_{80\%}$, five times in $Sat_{120\%}$, and only one time in $Sat_{160\%}$. Water stress was heavier in $Sat_{80\%}$, medium in $Sat_{120\%}$, and slight in $Sat_{160\%}$.

During the cropping period, the crop was not subjected to water stress in $Sat_{200\%}$. Rice was grown stress-free in $Sat_{200\%}$ while it was submitted to different stress levels in other treatments. Crops in $Sat_{80\%}$ stood longer under water stress while in $Sat_{160\%}$ and $Sat_{120\%}$ the time of stress was relatively shorter. In treatment $Sat_{80\%}$, water stress was observed at all growth stages, while significant stress was only observed from heading to harvest for $Sat_{160\%}$ and $Sat_{120\%}$. Applying water to 80%, 120%, and 160% of soil saturation induced severe, moderate, and low moisture stress during cropping season in $Sat_{80\%}$, $Sat_{120\%}$, and $Sat_{160\%}$ treatments, respectively, compared with farmer practice.

### 3.3. Growth Parameters

Growth parameters (plant height, tillers, and leaves number) were significantly affected by soil saturation levels at both active tillering and panicle initiation stages (Figure 3). Plant height (Figure 3A), tillers (Figure 3B), and leaves number (Figure 3C) decreased with the decreasing of soil moisture. The average decreases of 10%, 18%, and 14% were recorded for plant height, tillers, and leaves number, respectively, due to the reduction of supplementary irrigation.

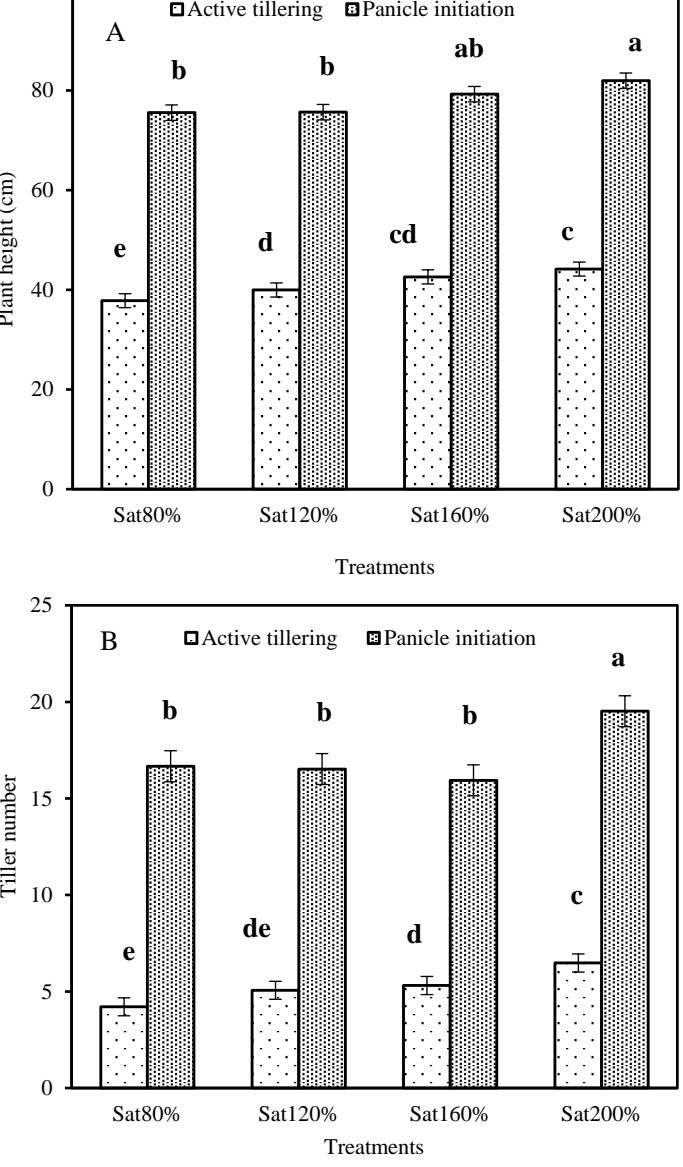

**Figure 3.** *Cont.*

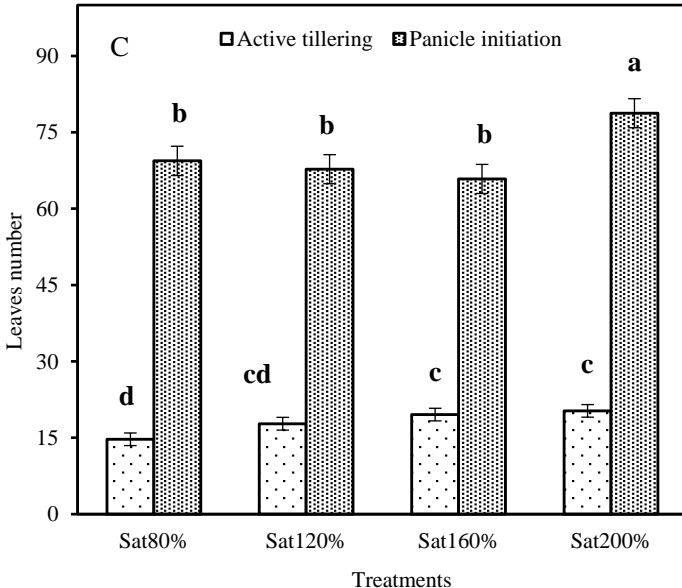

**Figure 3.** Comparison of the effect of treatments on plant height (**A**), leaves (**B**), and tiller (**C**) number.

### 3.4. Yield Components and Yield

At harvest, soil saturation decreases from 200% to 80% and has significantly affected yield components such as the tiller and panicle number per hill and panicle length (Table 1). The lowest number of tillers was registered in the lowest soil moisture treatment ($Sat_{80\%}$). However, $Sat_{120\%}$ and $Sat_{80\%}$ gave the same panicle number and length. Severe water stress mostly affected tiller and productive tiller numbers with a decrease of 13.08% whilst slow, moderate, and severe stress reduced panicle growth by 4.8%.

**Table 1.** Effect of treatments on tiller and panicle number and panicle length at harvest.

| Treatments | Tiller Number per Hill | Panicle Number per Hill | Panicle Length (cm) |
|---|---|---|---|
| $Sat_{200\%}$ | 14.83 ± 0.56 [ab] | 12.12 ± 0.48 [ab] | 26.76 ± 0.29 [a] |
| $Sat_{160\%}$ | 15.6 ± 0.61 [a] | 13.5 ± 0.56 [a] | 25.42 ± 0.27 [b] |
| $Sat_{120\%}$ | 15.1 ± 0.64 [a] | 12.04 ± 0.49 [b] | 25.42 ± 0.29 [b] |
| $Sat_{80\%}$ | 13.31 ± 0.50 [b] | 11.23 ± 0.43 [b] | 25.56 ± 0.27 [b] |
| *p* | 0.036 | 0.01 | 0.001 |
| Significance | * | ** | *** |

*** Means within columns not followed by the same letter are significantly different at $p < 0.001$ level by Duncan's test. ** Means within columns not followed by the same letter are significantly different at $p < 0.01$ level by Duncan's test. * Means within columns not followed by the same letter are significantly different at $p < 0.05$ level by Duncan's test; *p* Probability.

Panicle characteristics are shown in Table 2. Panicle weight, grain number, and grain weight per panicle were significantly affected by the reduction of soil saturation percentage. The highest values were recorded in higher water treatment ($Sat_{200\%}$), while $Sat_{160\%}$, $Sat_{120\%}$, and $Sat_{80\%}$ gave the same values. Slow, moderate, and severe water stress affected panicle characteristics with the same negative impact. Water stress has decreased panicle weight and grain weight by 17.82% and 19.47%, respectively.

**Table 2.** Water treatments effect on panicle weight, grain number, and grain weight.

| Treatments | Panicle Weight per Hill (g) | Grain Number per Panicle | Grain Weight per Panicle (g) |
|---|---|---|---|
| $Sat_{200\%}$ | 2.02 ± 0.08 [a] | 96.17 ± 5.55 [a] | 1.9 ± 0.12 [a] |
| $Sat_{160\%}$ | 1.64 ± 0.06 [b] | 76.77 ± 3.26 [b] | 1.43 ± 0.07 [b] |
| $Sat_{120\%}$ | 1.77 ± 0.07 [b] | 85.29 ± 5.07 [ab] | 1.57 ± 0.09 [b] |
| $Sat_{80\%}$ | 1.58 ± 0.05 [b] | 83.23 ± 3.54 [ab] | 1.59 ± 0.08 [b] |
| *p* | 0.000 | 0.02 | 0.005 |
| Significance | *** | * | ** |

*** Means within columns not followed by the same letter are significantly different at $p < 0.001$ level by Duncan's test. ** Means within columns not followed by the same letter are significantly different at $p < 0.01$ level by Duncan's test. * Means within columns not followed by the same letter are significantly different at $p < 0.05$ level by Duncan's test *p*; Probability.

Soil moisture decrease has significantly impacted 1000-grain weight, grain yield, and straw weight (Table 3). The values from different treatments showed that moisture decreases had the same effects (a reduction of 7.76%) on 1000-grain weight while $Sat_{200\%}$, $Sat_{160\%}$, and $Sat_{120\%}$ exhibited the same yield and straw weight. The lowest grain yield and straw weight was registered in lower soil moisture treatment. Slow and moderate moisture stress did not affect rice production, but severe stress decreased grain yield and straw weight by 19.97% and 26.09%, respectively.

**Table 3.** Soil saturation levels effect on 1000-grain weight, grain yield, and straw weight.

| Treatments | 1000-Grain Weight (g) | Grain Yield (kg/ha) | Straw Weight (kg/ha) |
|---|---|---|---|
| $Sat_{200\%}$ | 21.13 ± 0.21 [a] | 3316.67 ± 171.06 [a] | 3436.67 ± 143.79 [a] |
| $Sat_{160\%}$ | 19.79 ± 0.28 [b] | 2882.67 ± 174.01 [ab] | 3156.67 ± 119.83 [a] |
| $Sat_{120\%}$ | 19.47 ± 0.37 [b] | 3091.33 ± 181.06 [ab] | 3153.33 ± 133.47 [a] |
| $Sat_{80\%}$ | 19.2 ± 0.11 [b] | 2654.33 ± 134.70 [b] | 2540 ± 110.15 [b] |
| *p* | 0.000 | 0.036 | 0.000 |
| Significance | *** | * | *** |

*** Means within columns not followed by the same letter are significantly different at $p < 0.001$ level by Duncan's test. * Means within columns not followed by the same letter are significantly different at $p < 0.05$ level by Duncan's test; *p* Probability.

### 3.5. Water Productivity

Characteristics related to water use are presented in Table 4. The total seasonal amount of complementary irrigation used to reach different soil moisture treatments ranged from 3000 to 47,500 m³/ha. For irrigation water productivity, higher value was obtained in $Sat_{80\%}$ followed by $Sat_{120\%}$, while the lowest productivity was given by farmer watering practice. The highest rainwater productivity was recorded in farmer irrigation treatment. Decreasing supplementary water volume increased complementary irrigation water productivity by 92% and lowered rainwater productivity by 19%. Total water productivity was similar in $Sat_{80\%}$ and $Sat_{120\%}$. Complementary water reduction significantly increased irrigation water savings by 93.68%, 90.53%, and 87.37% in $Sat_{80\%}$, $Sat_{120\%}$, and $Sat_{160\%}$, respectively. Water-saving was higher in $Sat_{80\%}$, but the saving of one cubic meter of water in this treatment induced higher production losses compared with $Sat_{120\%}$. The best water-saving impact ($5 \times 10^{-3}$ kg/m³) was given by $Sat_{120\%}$ with good irrigation, rainwater, and total water productivities. Applying supplementary irrigation at 120% of soil saturation gave greater results.

**Table 4.** Effect of treatments on water productivity and water-saving.

| Treatments | Rainwater (m³/ha) | Complementary Irrigation Water (m³/ha) | Complementary Irrigation Water Productivity (kg/m³) | Rainwater Productivity (kg/m³) | Total Water Productivity (kg/m³) | Complementary Water Savings (%) | Complementary Water-Saving Impact (kg/m³) |
|---|---|---|---|---|---|---|---|
| $Sat_{200\%}$ | 3030 | 47,500 | $0.07 \pm 0.004$ [d] | $1.09 \pm 0.06$ [a] | $0.07 \pm 0.003$ [c] | | |
| $Sat_{160\%}$ | 3030 | 6000 | $0.48 \pm 0.02$ [c] | $0.95 \pm 0.06$ [ab] | $0.32 \pm 0.02$ [b] | $87.37 \pm 0.00$ [c] | $10 \times 10^{-3}$ |
| $Sat_{120\%}$ | 3030 | 4500 | $0.69 \pm 0.04$ [b] | $1.02 \pm 0.06$ [ab] | $0.41 \pm 0.02$ [a] | $90.53 \pm 0.00$ [b] | $5 \times 10^{-3}$ |
| $Sat_{80\%}$ | 3030 | 3000 | $0.88 \pm 0.04$ [a] | $0.88 \pm 0.04$ [b] | $0.44 \pm 0.02$ [a] | $93.68 \pm 0.00$ [a] | $15 \times 10^{-3}$ |
| *p* | - | - | 0.000 | 0.036 | 0.000 | 0.000 | - |
| Significance | - | - | *** | * | *** | *** | - |

*** Means within columns not followed by the same letter are significantly different at $p < 0.001$ level by Duncan's test. * Means within columns not followed by the same letter are significantly different at $p < 0.05$ level by Duncan's test; *p* Probability.

## 4. Discussion

In this study, saturated soil irrigation employed in a semi-arid rainfed rice farming system through the application of supplementary water depths is analyzed to make supplementary irrigation sustainable and beneficial by improving yield but also for saving more water for other purposes such as double cropping. Irrigation was applied during the weeks in regard to rainfall to get soil moisture close to saturation for a relatively long time by reducing water depths. Soil moisture trend was analyzed based on the stress limit line and the time of stress. Synergistic responses were observed between soil moisture trends, growth, yield components, and yield. Relatively high values of these parameters were recorded in the highest soil moisture treatment where crops were grown stress-free.

At the vegetative stage, highest height, tillers, and leaves numbers were achieved in stress-free water treatment ($Sat_{200\%}$), while the lowest value was given by the lowest moisture treatment ($Sat_{80\%}$) where rice was subjected to severe water stress. However, at panicle initiation the lowest soil moisture performed as well as moderate and low-stress treatments. A decrease of soil moisture from 200% to 80% of saturation at 40% interval, due to supplementary water restriction, retarded rice growth. Low, moderate, and severe water stress observed in $Sat_{160\%}$, $Sat_{120\%}$, and $Sat_{80\%}$, respectively, induced a decrease of height, tillers, and leaves number at different rates at the vegetative stage, but similar water stress impact was observed at panicle initiation. The highest growth decrease was caused by severe water stress while the lowest decreases were induced by low and moderate stress. It is well known that water stress reduces plants cells' water content and turgor. The water stress results in a decrease of cell enlargement leading to growth suspension or delay. Plant growth is lowered by severe drought stress due to a decrease in the stomata opening, which limits $CO_2$ uptake and hence reduces photosynthetic activity [31]. Stomata can completely close in mild to severe stress depending on stress intensity and timing. Under the water stress, cell expansion slows down or ceases, and plant growth is retarded.

At the early vegetative stage, severe stress highly retarded growth since young plantlets had not yet developed a stress adaptation mechanism to overcome water stress. Plants evolve mechanisms to adapt to water stresses during their growth. Plants' countermeasure to water stress usually starts after a certain time of stress. The time of developing physiological adaptation measures depends on the type of crop, time, and amplitude of stress. The reaction of the plants to water stress differs significantly depending upon the intensity and duration of stress as well as the development stage [32]. The impact of adaptation to water stress induced at the vegetative stage is mainly observed at panicle initiation.

Similar growth parameters obtained in severe, low, and moderate stress at panicle initiation may be explained by the recovery process in severe water stress treatment. Although no physiological measurement was done, plants under severe water stress may have slightly recovered from a relatively short time of severe stress and hence performed as well as those from moderate and low-stress treatments at panicle initiation. It is well known that water stress delays physiological development of plants as a result of tiller, height, and leaves reduction. Under water stress, a rice plant reduces evapotranspiration, which inversely reduces growth parameters such as height, tillers, and leaves number. Water stress significantly reduced the tillers and height of all rice cultivars [33]. Water availability is the key driver of sustainable crop production [34], and its stress adversely affects the physiology of plants [35]. However, subjected to severe water stress, a plant develops adaption measures that involve physiological processes for partial and full recovery depending on stress duration. Plants evolve physiological adaptations to adapt to water stresses [36,37]. One of the most important adaptation measures for plant recovery is the development of a roots system. Roots are the main organ for water and nutrients uptake from the soil and the most important part in the response of the plant to drought stress [38]. Under severe water stress, the plant may have developed a large and deep root system to faster extract water inducing growth recovery in $Sat_{80\%}$. Under a certain level of water stress, plants react by adapting in order to overcome drought by extracting water faster in depth through the development of a large roots system [26]. The recovery extents depend on pre-drought intensity and duration [39]. Kima et al. (2015) [18] mentioned that plants slowly recovered from high

water stress at the vegetative stage. Water stress in $Sat_{160\%}$ and $Sat_{120\%}$ was not sufficient to stimulate plant development.

Although there was a slight recovery in $Sat_{80\%}$, long duration of stress from the end of the initial stage to panicle initiation may have stopped the recovery process, and complementary irrigation reduction decreased rice growth at panicle initiation compared with farmer water stress-free irrigation. Water stress induced by the decrease of soil moisture reduced plant height, tillers, and leaves' development rate. It is well known that under water stress a plant reduces physiological activities that lead to a delay or suspension of the development of some parameters such as height, tillers, and leaves number. Severe and prolonged water stress may damage the structure of the photosynthetic, resulting in lower plant height [40]. Lilley and Fukai (1994) [41] showed that water deficit reduced the rate of apical development.

Water stress levels induced at vegetative, reproductive, and repining stages differently affected yield components at harvest. Severe water stress highly impacted tiller number, while reproductive tillers (panicle number per hill) were affected by both moderate and severe water stress. Slight water stress induced during the farming period did not have a significant impact on tillers' panicles at harvest. Panicles were more affected by relatively sharp water stress. The effect of soil moisture decline on tillers and panicles may be explained by the severity of the stress and the sensitivity of the rice crop at a specific stage. At the reproductive stage, where panicles are set, water availability is crucial so that severe and moderate water stress decreases panicle number per hill. Water stress at that stage may have decreased the heading rate, induced the sterility of flowers, increased the number of unproductive tillers, and shorted panicles. However, grain number per panicle was not affected by any stress. Headed panicles have overcome water stress and statistically produced the same number of grain. Nevertheless, panicle characteristics (panicle weight and grain weight per panicle) and 1000-grains were significantly affected by any amplitude of stress. Panicles were sensitive to water stress. Drought at the seedling and tillering stages resulted in zero panicles for all evaluated rice cultivars [42]. At the grain formation stage, where grain should be filled, any water stress may drastically inhibit, delay, or suspend the filling rate. Although grain filling was not measured in our study, low, moderate, and severe water stress may have delayed filled grain ratio resulting in panicle weight, grain weight per panicle, and 1000-grain yield.

Despite variable impacts of soil saturation levels on yield components, grain yield and straw weight were only affected by severe water stress. Statistically, stress-free, low stress, and moderate stress treatments gave similar yields and straw weight. Severe soil moisture stress decreased grain yield and straw weight, while slight stress did not have any impact on harvest. Since the rice crop was subjected to heavier water stress at the vegetative stage and the slight recovery process observed at panicle initiation may have stopped due to the severity and the time of stress, severe water stress highly affected yield. A relatively long time and high amplitude of stress in the lowest soil saturation drastically decrease yield. Since grain number and grain weight did not vary in $Sat_{160\%}$, $Sat_{120\%}$, and $Sat_{80\%}$, the grain filling rate may explain yield trends in these treatments. Soil moisture decrease due to severe exposure and extended water stress in $Sat_{80\%}$ from the vegetative stage to harvest may have decreased filled grain percentage, which induced the lowest yield in $Sat_{80\%}$. Plant development and yield were highly influenced by drought [42]. Sarvestani et al. (2008) [33] reported that a reduction in yield due to water stress resulted from the reduction of fertile panicle and filled grain percentage. On the contrary, under low and moderate stress, gross yield and straw weight were not affected because of low-stress amplitude. Probably, the rice may have recovered yield from slight water stress.

The increasing of soil moisture increased water productivity up to optimal saturation levels followed by a productivity decrease due to the use of a large amount of complementary irrigation whilst a slight yield increase and even a decrease were observed. Applying high ponded water depth to reach 200% of soil saturation may have increased hydraulic head pressure, which induced high water losses through percolation. Generally, when irrigation water depth increases, percolation increases with nutrients losses through leaching. The increase of the infiltration rate led to a mismatch between

plant water absorption and irrigation application, which induces low nutrients uptake. That led to low irrigation water productivity in farmer irrigation practice and $Sat_{160\%}$ treatments. Zain et al. (2014) [43] confirmed that rice yield was mostly influenced by water use efficiency. Regarding complementary irrigation (0.69 kg/m$^3$), rainwater (1.02 kg/m$^3$), and total water (0.41 kg/m$^3$) productivities with water savings (90.53%) and their impacts ($5 \times 10^{-3}$ kg/m$^3$), saturated soil irrigation at 120% appeared suitable. Production losses were two and three times lower in this treatment compared with $Sat_{160\%}$ and $Sat_{80\%}$, respectively. The production lost by saving one unit of water is a relevant indicator for identifying suitable irrigation water [26]. Since the lowest production lost by saving one cubic meter of water was observed in $Sat_{120\%}$, it should be chosen as the adapted soil saturated irrigation.

## 5. Conclusions

In this study, we compared soil saturation levels including farmers' watering practice to identify suitable saturated soil irrigation for increasing both yield and water productivity. Increasing supplementary water increased rainfed rice growth and yield up to optimum levels, after which additional water becomes unproductive. Water stress at the early vegetative stage decreased growth parameters at panicle initiation, but recovery was observed at harvest depending on the severity of stress. The rice crop recovered from low and moderate moisture stress and yielded as well as that of the stress-free farmer irrigation practice.

Complementary water depths were field evaluated for only one rainy season. This can be seen as a limitation of the study, but it was revealed that applying supplementary irrigation at 7 days interval to get soil saturated at 120% yielded the best results in terms of yield and water productivity. Saturated soil irrigation can be employed in a semi-arid rainfed rice farming system as an effective measure to overcome drought spells and rainwater shortage and to save more water for other purposes. The technique seems adapted to dry climates and can be recommended to replace the stress-free complementary irrigation practiced by farmers. The experiment is going to be replicated for two other consecutive rainy seasons to better consider rainfall variability and add other physiological analyses such photosynthesis, roots development, heading rate, and leaves' chlorophyll.

We advocate that saturated soil irrigation can be employed in supplementary irrigation in the semi-arid rainfed rice system as a suitable drought alleviation measure by adjusting complementary water depth and irrigation intervals with respect to climate conditions in order to match the high water demand period of a rice crop with heavy rainfall occurrence.

**Author Contributions:** A.S.K. initiated the study's conceptual protocol and conducted the field experiment. P.A.W.O. collected the data in the field. E.K. assisted with the data collection and quality check. B.B. contributed to the design of the experimental protocol and also to the statistical discussion. O.T., S.T., and H.N. assisted the first author in editing the manuscript. W.-G.C. reviewed the first draft of the manuscript and suggested statistical analysis. Y.-M.W. was the principal advisor of the first author and helped to finalize the paper. All authors have read and agreed to the published version of the manuscript.

**Funding:** This research received no external funding.

**Acknowledgments:** The authors would like to thank the Agricultural Research Institute of Burkina Faso (INERA) for hosting and providing the facilities for the study. Acknowledgments go to the SAPEP project for funding this research; without their support, this study would not have been possible.

**Conflicts of Interest:** The authors declare no conflict of interest.

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
