# Peer review of "Evaluating Supplementary Water Methodology with Saturated Soil Irrigation for Yield and Water Productivity Improvement in Semi-Arid Rainfed Rice System, Burkina Faso"

_sustainability, doi:10.3390/su12124819_

Round 1

Reviewer 1 Report

Optimizing the crop production is a commonly known challenge in all countries but in the semi-arid/arid ones it is a priority. People existence is based on enough yield. From this point of view, in my opinion, the scope of presented article is worthy for investigation. The paper is prepared very clearly, field experiments are described in details, course of investigation/results/conclusion is logic. It should be highlighted that Authors connect science research with practice. Obtained result could be implement to minimize wasting the water with highest possibility of rice productivity. In my opinion the paper could be published. I found only one little mistake on Figure 3 - in the word "Panicle" the letter "e" is missed.

Reviewer 2 Report

Dear Authors,

I reviewed the manuscript entitled "Evaluating Supplementary Water Methodology with Saturated Soil Irrigation for Yield and Water Productivity Improvement in Semi-Arid Rainfed Rice system, Burkina Faso" and my impression, at this stage, is not entirely positive. So, several improvements are needed before the manuscript can be published. For this, some suggestions to improve the manuscript are noted in the attached copy.

Overall, I think that the manuscript deal with an interesting main topic for Burkina Faso regions, because is expected that water deficit will affect agricultural crops due to climate changes. Specifically, the study aim to determine a suitable irrigation water depth that may improve supplementary irrigation water use efficiency through saturated soil irrigation (SSI) technique, which could increase water productivity and farm output in dry climate conditions of Burkina Faso region.

However, several shortcomings have been highlighted that should be addressed.

In my opinion, at least two or three main issues have to be adequately addressed and improved (please see the attached file for a complete view).

1) the irrigation volumes were calculated according with soil properties of undisturbed soil but a soil tillage was carried out;

2) the discussion was mainly developed on conjectures because, as stated by Authors (R.328), "...no physiological measurement was done...". Therefore, references of literature should be provided to support the hypotheses made.

I believe that the authors will be able to improve the manuscript and I can't wait to evaluate the improvements made.

Best reegards

Reviewer

Reviewer 3 Report

It is a good job but I recommend that the authors dept into the methodology and the results. In the introduction they must clearly indicate the novelty of it. Failing that, they should enhance the applicability in similar arid areas and its importance.

The author must differentiate what is “an experimental work: experimentation” from an “investigation”, with long-term objectives.

There are specific recommendations for each section in the attachment.

Round 2

Reviewer 2 Report

Dear Authors,
it might seem that an answer has been given to the questions posed, but in reality I don't see substantial changes to the original version, as the discussion of the results has remained unchanged in substance.

Specifically, although some (few) references of literature were provided, inserting only the progressive number whitin brackets is not equivalent to explaining what is the relevance of the reference, and what similarities exist between the latter and the research carried out. I would have expected a more thorough review and, from my point of view, the request remains unsatisfied. Also, diagrams of Figures 2 have not still adequately commented, because the explanation of the temporal dynamics remains weak and not very explanatory.

Therefore, since my questions have not yet found an adequate answer, they must be re-proposed again, hoping for a positive outcome aimed at improving the manuscript.

Reviewer 3 Report

I think the work is practical and applicable. Congratulations

General

Line 64-73 should be reinforced with bibliography

Line 97-103 should be reinforced with bibliography

Delete general equations and expressions known to all (1, 2, 3, 4, 5, 6, 7) from the text. It is enough with the reference to the author who proposed it.

In Figure 2, reducing the y-axis interval (water fluctuation) improves the vision of the results. e.g. (-5 to -28)

Figure 3 in an appendix or supplementary material. Your information is expressed in the tables.

Round 3

Reviewer 2 Report

Dear Editor,

I am pleased to note that the authors have made an effort to improve the manuscript by improving it. Therefore, I believe it can now be accepted for publication on Sustainability.

Best regards

MC